# Engagement in research capability-building: Impact on healthcare workforce attraction and retention in rural and remote Australia – A scoping review protocol

Tsegaye G. Haile[1,2]*, Hilary Wallace[3], Justin Manuel[3], Mohamed Estai[3]*

**1** Curtin School of Population Health, Curtin University, Perth, Western Australia, Australia, **2** Department of Health Systems and Policy, Institute of Public Health, University of Gondar, Gondar, Ethiopia, **3** WA Country Health Service, Bentley, Western Australia, Australia

* t.haile2@postgrad.curtin.edu.au (TGH); mohamed.estai2@health.wa.gov.au (ME)

## Abstract

### Background

Geographic maldistribution of the health workforce remains a major challenge in Australia, with rural and remote communities experiencing persistent shortages that undermine access to and quality of care. Engagement in research is recognised as a potential mechanism for professional growth, continued learning, and improved workplace environments. Providing opportunities for health workers to participate in research or research capability-building (RCB) may therefore support workforce outcomes such as attraction and retention. This scoping review will identify and map existing evidence on the relationship between research engagement and health workforce outcomes in rural and remote Australia, and to summarise the factors that contribute to successful implementation of such initiatives.

### Methods and analysis

We will conduct a scoping review of published and grey literature from Australia (2000 to present). Searches will be undertaken in CINAHL (EBSCO), Embase (Ovid), Global Health (Ovid), MEDLINE (Ovid), and PubMed, following the PRISMA-ScR 2020 guidelines and the Joanna Briggs Institute methodology for scoping reviews. Additional searches will be conducted through Google, Google Scholar, organisational websites, and snowballing of reference lists from included studies. Search terms will address four core concepts: (i) health professionals; (ii) research and RCB; (iii) workforce outcomes; and (iv) rural and remote Australian settings. Both qualitative and quantitative evidence will be included. Data will be synthesised using descriptive and thematic analysis, combining deductive approaches informed by the socioecological model and inductive approaches. Subgroup analyses will be undertaken where appropriate to provide deeper insights into the findings.

**Data availability statement:** No datasets were generated or analysed during the current study. All relevant data from this study will be made available upon study completion.

**Funding:** The author(s) received no specific funding for this work.

**Competing interests:** The authors have declared that no competing interests exist.

## Ethics and dissemination

This scoping review will not involve human participants or primary data collection and does not require ethical approval. The results of this review will be published in a peer-reviewed journal.

## Background

Health workforce shortages in remote and rural Australia significantly hinder access to care, leading to poorer health outcomes and increased healthcare costs [1–3]. The latest report from the Australian Institute of Health and Welfare (AIHW) indicates that people living in remote areas have poorer access to health services than those in major cities [4]. In 2022, the full-time equivalent (FTE) rate of clinical health professionals in remote and very remote areas is 1,938 and 1,846 per 100,000 population, respectively, compared with 2,248 per 100,000 in major cities, highlighting a notable maldistribution [4]. This disparity contributes to a higher reliance on patient transfers, contract medical officers, and agency staffing [1]. Delays in care, reduced availability of specialist services, and evidence from other low-resource settings further suggest that inadequate health workforce capacity undermines service provision and compromises the quality and equity of healthcare [1,5,6].

The maldistribution of the health workforce, along with weak coordination between workforce training and employment in remote areas, exacerbates existing shortages [7]. Increased staffing of rural clinics in response to targeted funding is often temporary due to unstable funding, limited job security, and reluctance among clinicians to commit to rural practice [8]. In Western Australia, many remote clinics rely on fly-in fly-out (FIFO) models, with rotating clinicians providing intermittent care [9,10]. While offering short-term relief, this approach undermines continuity, leading to poorer patient outcomes and increased strain on emergency services.

National reports also indicate that rural clinical roles are often perceived as career-limiting, largely due to limited opportunities for specialisation, mentorship, and career progression [11,12]. For example, research has shown that such perceptions can influence medical graduates' willingness to consider rural practice, particularly among those seeking highly specialised careers or structured career pathways [13,14]. These systemic barriers contribute to ongoing workforce instability, further hindering effective health service delivery in rural and remote communities. This underscores the need to address career development pathways in rural health workforce planning.

Beyond addressing workforce shortages and distribution to improve the quality of rural healthcare, there is growing recognition that building workforce research capability is essential to improving rural health outcomes. One promising approach involves developing practitioner–researcher pathways through research capacity-buildings (RCB).

RCB involves strengthening the skills, systems, resources, and organisational structures that enable individuals and institutions to generate, interpret, and apply

research to improve practice and decision-making [15]. These RCBs aim to support rural and remote clinicians to develop research skills, generate locally relevant evidence, and contribute to health service improvement in rural and remote areas. For instance, the Rural Research Capacity Building Program (RRCBP) in New South Wales, initiated in 2006, aims to increase the number of rural and remote health workers equipped with evaluation and research skills [16,17]. Participants report improvements in service delivery, patient outcomes, and professional confidence. Other research capacity-building programs in Australia include the Research Ready Grant Program in Queensland [18] and the Supporting Translation of Research in Rural and Remote health settings (STaRR) program in Victoria [19,20]. These programs contribute to professional development, inform rural health service delivery models, and support evidence-based policymaking.

Despite these gains, rural practitioners continue to face structural and contextual barriers to conducting research. A study by Wong et al. identified macro-level challenges such as limited rural research funding and workforce policy misalignment; meso-level factors including inadequate organisational prioritisation of research; and micro-level issues such as the need for researchers to understand local sociocultural dynamics [21]. These barriers can limit the sustainability and impact of RCB efforts in rural contexts. Thus, to inform remote workforce development strategies in Australia, a scoping review will be undertaken to examine how engagement in research and participation in RCB influences health workforce outcomes such as retention and attraction in rural and remote settings.

## Research questions

Our scoping review will address the following questions: (a) what is the current evidence on the association between health workforce engagement in research and/or RCB and attraction and retention in rural and remote Australia? and, (b) what enablers and barriers influence RCB initiatives to enhance health workforce outcomes in rural and remote regions of Australia?

## Objectives

In our scoping review, we will:

- Identify and map evidence on the relationship between RCB and workforce attraction and retention in rural and remote Australia.

- Identify enablers and barriers to implementing RCB initiatives that enhance workforce outcomes in these settings.

- Map effective strategies to strengthen future RCB efforts for improving workforce outcomes.

## Methods

### Protocol design

We will undertake a scoping review of the current evidence, including both published and grey literature, examining the relationship between health workforce engagement in research (including through RCB programs) and outcomes such as attraction and retention in rural and remote settings. Our review will follow the Preferred Reporting Items for Systematic reviews and Meta-Analyses Extension for scoping Reviews (PRISMA-ScR) 2020 guidelines [22] and will be conducted according to the Joanna Briggs Institute (JBI) [23] methodology for scoping reviews.

### Data sources and search strategy

We will search major databases, such as CINAHL (EBSCO), Global Health (OVID), Medline (OVID), Scopus and ProQuest Central, and PubMed, to identify relevant studies. We will also search google, Google scholar and organisational websites such as AIHW, Australian Health Practitioner Regulation Agency, Rural Health Workforce Australia, and

Australian Health Practitioner Regulation Agency, Health Education and Training Institute, and Western Research Academic Health Science Centre, to identify grey literature. Organisational websites included in the Western Australia Country Health Service TTR Report "Doing More Research Well" (Wallace 2022) will also be included. Moreover, additional relevant studies identified within the included articles will be assessed and incorporated (through citation snowballing) into the final review.

We used the population, concepts, and context (PCC) approach to formulate the review questions [24]. Accordingly, the population will include all health professionals including the four professional categories: nurses and midwives, medical practitioners, dental practitioners, and allied health according to the AIHW Health workforce report [4]. The concept (phenomena of interest) is engagement in research or capability building and its effect on workforce attraction, retention and/ or stability. Lastly, the context includes remote and rural areas of Australia which could consists of remote areas, rural areas, disadvantaged areas, underserved areas, and or far to reach areas.

Thus, our search terms use the border concepts of i) healthcare workforce, ii) engagement in research and research capability budling initiatives/programs/strategies, iii) health professionals' attraction and retention, and iv) settings (rural and remote Australia) to build the searching strategies (S1 Table).

### Study selection

The study selection process will follow the PRISMA-ScR guidelines. All records retrieved from the search will initially be imported into Covidence™ for management. Duplicate entries will be removed automatically and manually checked. Before formal screening, two reviewers (TGH and ME) will complete a calibration exercise on a random sample of 25–50 titles and abstracts using the draft eligibility criteria. Discrepancies will be discussed to refine the screening instructions. Following the discussion, first, one reviewer (TGH) will screen all titles and abstracts against the selection criteria, while a second reviewer (ME) will independently screen a random sample comprising 25% of the records. Discrepancies and any records marked as 'maybe' will be discussed and resolved by both reviewers (TGH and ME). Full texts will then be retrieved for all studies marked as "include" or "uncertain." Screening decisions at each stage will be documented and reported using a PRISMA-ScR flow diagram. Next, two reviewers (TGH and ME) will independently assess all retrieved full-text articles against the predefined inclusion and exclusion criteria. Reasons for exclusion will be recorded in Covidence™ using standard categories to ensure transparency. Any disagreements regarding study eligibility will be resolved through discussion or, if required, consultation with a third reviewer. To enhance completeness, backward and forward citation tracking (snowballing) will be conducted on all included studies. Newly identified studies will undergo the same eligibility assessment procedures as the primary set of records.

### Eligibility criteria

We will include all primary studies and reports published since 2000, including both qualitative and quantitative studies, reports such as governmental/organisational reports and grey literature. We will include studies conducted in both metro and rural areas and extraction for rural results will be done separately. We will exclude systematic reviews (though their references will be used for snowballing) as well as studies conducted exclusively in urban areas and major cities. The detailed inclusion and exclusion criteria are presented in Table 1.

### Data extraction and analysis

Data from all eligible studies will be extracted using a structured Excel™ spreadsheet developed for this review. Two reviewers (TH and ME) will independently extract data from an initial sample of five studies to pilot and refine the extraction tool. Any discrepancies will be discussed, and the tool will be adapted to ensure it captures all required study characteristics and reporting elements. Following this pilot phase, one co-author (TH) will extract data from the remaining studies, and a second reviewer (ME) will verify the extracted information for accuracy and completeness.

**Table 1. Study inclusion and exclusion criteria.**

| Criteria | Definition | Inclusion | Exclusion |
|---|---|---|---|
| Population | • Health professionals | All health professionals according to the Australian Health and welfare health workforce classifications including<br>◦ Nurses and midwives<br>◦ Medical practitioners<br>◦ Dental practitioners<br>◦ Allied health | • Studies conducted among non-registered health professionals and students |
| Concept/interventions | • Research capability-building | ◦ Engagement in research and research capability building initiatives<br>◦ Research capacity building<br>◦ Research development<br>◦ Research mentoring and supervision<br>◦ Research supportive incentives | • Not involved/engaged in research or research capability building initiatives//programs<br>• Other professional development programs |
| Outcome | • Health workforce outcomes | ◦ Attraction and retention<br>◦ Satisfaction<br>◦ Stability<br>◦ Intention to stay/leave | |
| Type of studies | • Types of articles/evidence to be included in the review | ◦ All observational studies; published and publicly available grey literatures<br>◦ Qualitive, quantitative and mixed method studies | • Studies not relevant to the review questions or missed one or two of the eligibility criteria<br>• Letters and editorials<br>• Conference papers |
| Context/settings | • Rural and remote settings | ◦ Regional<br>◦ Remote<br>◦ Rural<br>◦ Underserved areas<br>◦ Hard to reach areas<br>◦ Countryside<br>◦ Non-metropolitan<br>◦ Studies with both rural and metro will be included and results for rural will be extracted. | • Urban areas<br>• Major cities<br>• Metropolitan areas/cities |
| Country | | ◦ Studies conducted in Australia | • Studies conducted outside Australia |
| Limiters | Year of publications | ◦ From 2000 onwards | • Before 2000 |
| | Studies species | ◦ Human studies | • Animal studies |

The abstraction sheet is drafted for both qualitative and quantitative evidence. Key information to be collected will include: 1) study characteristics (author, year of publication, country/setting), study design (qualitative, cross sectional, interventional, cohort, case-control), sample size, data collection or methods of analysis); 2) participants characteristics (types of health workforces, speciality, year of experience, and workplace or service context); 3) characteristics of RCB (type of program, area of research, duration or intensity of engagement), and 4) reported outcomes, including workforce retention, recruitment or attraction, sustainability, turnover, intention to leave, and other relevant effects (Table 2). A separate extraction sheet will be used for grey literature sources, capturing document characteristics and key findings.

A descriptive qualitative analysis will be conducted using both deductive and inductive thematic approaches. The deductive analysis will apply the socioecological model to organise the impacts of RCB initiatives across individual, team, organisational, and system levels (Fig 1). In parallel, an inductive analysis will be undertaken, first to identify the key enablers and barriers influencing the implementation and effectiveness of RCB activities, and second to map the pathways through which RCB contributes to workforce outcomes, including its interactions with broader retention and workforce-strengthening strategies. Themes will be generated from the extracted data and refined iteratively, with quantitative findings incorporated to support or extend the qualitative insights. In line with JBI guidance for scoping reviews, we

**Table 2. Data abstraction sheet to describe the study in the final review.**

**Data elements**

| | Quantitative | Qualitative |
|---|---|---|
| Study characteristics | • Author(s), year<br>• Year of study<br>• Study area<br>• Study design (prospective cohort, retrospective cohort, case-control, and cross sectional) | • Author(s), year<br>• Year of study<br>• Study area<br>• Study design (phenomenological, case study, ground theory, iterative, …) |
| Study participants | • Study population/participant (types of health professionals)<br>• Study population/participants' details (mean/median age, proportion by profession, department, and health facility types, sample size, ….) | • Study population/participant (types of health professionals)<br>• Study population/participants' details (proportion by profession, department, and health facility types, sample size, ….) |
| Description of program/ initiatives | • Types of interventions (research engagement, research capability building initiatives, mentoring, supervision …)<br>• Areas of research<br>• Length of research experience<br>• Purpose of research engagement/involvement | • Types of interventions (research engagement, RCB initiatives, mentoring, supervision …)<br>• Areas of research<br>• Length of research experience<br>• Purpose of research engagement/involvement |
| Methods and analysis | • Method of data collection<br>• Analysis techniques with their effect measurements | • Method of data collection<br>• Analysis techniques with their measurements |
| Outcomes | • Attraction<br>• Retention<br>• Intention to stay/leave<br>• Sustainability<br>• Stability<br>• Turnover rate | • Attraction<br>• Retention<br>• Intention to stay/leave<br>• Sustainability<br>• Stability<br>• Turnover rate |
| Key findings | • Main results (effect estimation)<br>• Key factors | • Key findings identified from the qualitive analysis<br>• Many themes/areas of findings<br>• Barriers and facilitators of RCB initiatives' implementation |

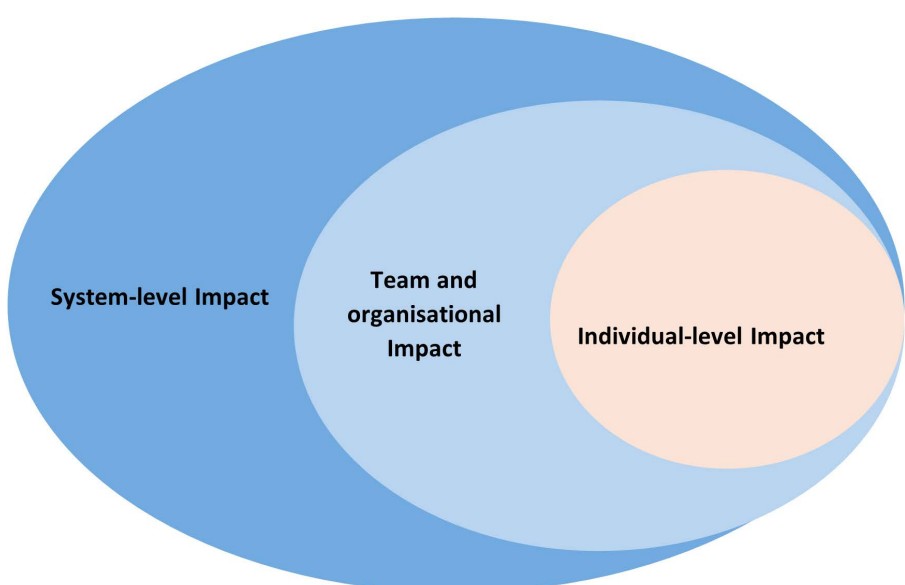

**Fig 1. Impact of RCB on different levels guided by Socioecological Framework.** *A framework which will be used to present the impact of RCB at different levels based on the socioecological model. RCB: Research Capacity Building.*

will not appraise study quality or risk of bias, as the purpose of this review is to map and characterise the existing evidence base rather than to synthesise effect estimates.

## Patient and public involvement

Patients or the public were not involved in the design, or conduct, or reporting, or dissemination plans of our research.

## Ethics and dissemination

Ethical approval is not necessary for this study, as it will be a retrospective review of publicly available evidence sources and does not collect primary data. The findings of the scoping review will be disseminated through publications in peer-reviewed journals and presentations at symposia and conferences. To ensure that the review findings reach relevant stakeholders, a dissemination strategy will be developed later in the review process.

## Study period and amendments of review

Our review will include studies published from January 1, 2000, to the date of the final search. Any significant protocol amendments will be documented and reported with the final scoping review.

## Discussion

Our scoping review will address gaps in current knowledge by exploring evidence on how health workforce engagement in RCB influences outcomes such as attraction, retention, professional development, job satisfaction, and career progression in rural and remote areas Australia, settings where workforce shortages are persistent and well recognised. By integrating both peer-reviewed and publicly available unpublished literature, and synthesising qualitative and quantitative evidence, the scoping review will map the breadth and depth of available evidence, identify enablers and barriers to implementation, and highlight strategies proposed to strengthen the effectiveness of RCB programs. Ultimately, our scoping review findings will provide insights into how RCB can support professional growth, foster a sense of purpose and connectedness, and ultimately contribute to workforce stability in underserved settings.

The extent to which RCB affects workforce outcomes is likely shaped by contextual factors, including health service type, staff experience, disciplinary background, resource availability, organisational culture, and the systems in place to support research activity. Differences in infrastructure, mentorship, feedback mechanisms, and perceived benefits may all influence engagement. Our scoping review will therefore explore these contextual factors or constrains successful implementation in rural and remote contexts, with the aim of informing future policy and practice.

Our scoping review will also position RCB within the broader suite of interventions designed to strengthen the rural and remote health workforce. We will examine how capability-building initiatives intersect with complementary strategies, such as professional development programs, mentoring and supervision schemes, financial incentives, and organisational retention efforts. By identifying co-interventions, potential synergies, and areas of duplication or missed opportunity, the review will highlight best practices and generate evidence-informed recommendations to enhance attraction, retention, and workforce sustainability at individual, team, organisational, and system levels.

This scoping review will apply a rigorous and transparent methodology, guided by the PRISMA-ScR framework and the JBI approach. Independent, duplicate screening of titles, abstracts, full texts, and data extraction will enhance reliability and minimise reviewer bias. The use of a comprehensive search strategy, covering multiple bibliographic databases, grey literature sources, and organisational repositories, will reduce publication bias and ensure wide capture of relevant evidence. Predefined eligibility criteria and a structured data extraction process will enable systematic mapping of how research RCB initiatives influence health workforce outcomes in rural and remote Australian settings.

Some limitations are anticipated. The inclusion of grey literature may introduce variability in methodological quality, as such works does not undergo peer review; however, this is consistent with the aim of our scoping review, which seeks to

map the scope of evidence rather than assess its robustness. Considerable heterogeneity across study designs may limit the comparability of findings, and variability in definitions of 'rural' and 'remote' may change the synthesis. Challenges in identifying unpublished research and CBP activities may also lead to under-representation of some initiatives.

Despite these anticipated limitations, our scoping review will offer important insights for policymakers, health service leaders, and research administrators. By clarifying how research engagement and RCB operates within rural and remote settings, and the conditions under which they are most effective; this work will support for the design of future programs, inform workforce policy, and guide targeted investment in research capacity as a lever for improving workforce stability.

## Supporting information

**S1 Table. Search strategy for a scoping review of engagement in research capability-building and its impact on healthcare workforce attraction and retention in rural and remote Australia.**
(DOCX)

## Acknowledgments

We would like to acknowledge Vanessa Varis (Faculty of Health Sciences librarian) for her assistance with the search strategy, databases, and the use of COVIDENCE.

## Author contributions

**Conceptualization:** Tsegaye G. Haile, Hilary Wallace, Justin Manuel.

**Data curation:** Tsegaye G. Haile.

**Formal analysis:** Tsegaye G. Haile.

**Investigation:** Tsegaye G. Haile, Hilary Wallace, Justin Manuel, Mohamed Estai.

**Methodology:** Tsegaye G. Haile, Hilary Wallace, Justin Manuel, Mohamed Estai.

**Project administration:** Tsegaye G. Haile, Justin Manuel, Mohamed Estai.

**Software:** Tsegaye G. Haile.

**Supervision:** Hilary Wallace, Justin Manuel, Mohamed Estai.

**Validation:** Tsegaye G. Haile, Hilary Wallace, Justin Manuel, Mohamed Estai.

**Visualization:** Tsegaye G. Haile, Hilary Wallace, Mohamed Estai.

**Writing – original draft:** Tsegaye G. Haile.

**Writing – review & editing:** Tsegaye G. Haile, Hilary Wallace, Justin Manuel, Mohamed Estai.

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
