## [Decision Letter · Decision Letter 0]

22 Oct 2025

Dear Dr. Haile,

Both reviewers recognise the importance of exploring how research engagement and capability-building may support recruitment and retention of the rural and regional health workforce. Nevertheless, they highlight that the manuscript requires clearer conceptual framing, justification of scope, and refinement of methods to ensure transparency and reproducibility.

I concur with the reviewers that the **scoping review approach** is appropriate for this exploratory topic and should be explicitly and consistently presented throughout the manuscript. Below, I summarise the required and recommended revisions.

Clarify and align the review type.Justify the scopeDefine key terms and maintain consistencyStrengthen the rationale and theory of changeIntegrate existing evidence and program outcomes.Provide detailed search strategy and feasibility statement.Minor textual corrections.

We look forward to receiving your revised manuscript.

Kind regards,

Muhammad Shahzad Aslam, Ph.D.,M.Phil., Pharm-D

Academic Editor

PLOS ONE

2.. If the reviewer comments include a recommendation to cite specific previously published works, please review and evaluate these publications to determine whether they are relevant and should be cited. There is no requirement to cite these works unless the editor has indicated otherwise.

Additional Editor Comments:

1-The manuscript and title indicate a scoping review protocol (appropriate given the likely heterogeneity and exploratory state of the evidence). Ensure all instances in the text, abstract, and methods consistently describe a scoping (not systematic) review and that the rationale follows PRISMA-ScR logic (i.e., mapping concepts, clarifying definitions, identifying gaps, not pooling/comparing effects).

2-Expand to include OECD/high-income rural/remote contexts (with Australia as a prespecified subgroup) to enhance yield and generalisability; or Retain Australia-only but explicitly justify (policy relevance, programmatic uniqueness, feasibility) and discuss implications for transferability.

3-Define your concept of interest (research engagement/capability-building)

4-State how you will contextualise findings alongside other known strategies in the discussion (e.g., mapping co-interventions, noting where research is embedded within larger retention packages).

5-Use one set of geographic terms throughout. If Australia-specific classifications are needed, define at first use.

6-Specify Population–Concept–Context

7-Provide a transparent draft search string (per database) and list of key terms/synonyms

8-Elaborate and explain in detail Study selection & calibration

9-Describe narrative mapping and tabulation, grouping by programme type, profession, and context; depict the programme theory as a diagram; identify evidence gaps and future research needs. No meta-analysis is expected.

10-Several language issue raised by reviewer. Addressed all reviewer comments.

Reviewers' comments:

Reviewer's Responses to Questions

**Comments to the Author**

1. Does the manuscript provide a valid rationale for the proposed study, with clearly identified and justified research questions?

Reviewer #1: Partly

Reviewer #2: Partly

2. Is the protocol technically sound and planned in a manner that will lead to a meaningful outcome and allow testing the stated hypotheses?

Reviewer #1: Yes

Reviewer #2: Partly

3. Is the methodology feasible and described in sufficient detail to allow the work to be replicable?

Reviewer #1: Yes

Reviewer #2: Yes

4. Have the authors described where all data underlying the findings will be made available when the study is complete?

Reviewer #1: Yes

Reviewer #2: Yes

5. Is the manuscript presented in an intelligible fashion and written in standard English?

Reviewer #1: Yes

Reviewer #2: Yes

You may also provide optional suggestions and comments to authors that they might find helpful in planning their study.

Reviewer #1: Workforce attraction: I suggest workforce recruitment unless workforce attraction is the necessary specific term.

Research capability building sounds a bit clunky, if this is not an already used term then I suggest research skill development. If it is the accepted term, please ignore.

Spell out acronyms the first time you use them like FTE in the first paragraph.

Please use consistent terminology. Currently there is: remote and very remote, rural and regional communities, and rural and remote. If these specific terms are used for a reason, please define.

In the intro, line 63 please use a phrase like “for example” when noting that research shows that medical grads are less willing to consider rural practice since the paper is inclusive to all clinicians, not just physicians.

Are there reported outcomes for individuals who have participated in RRCBP or STaRR in terms of career development, satisfaction of participant’s career, rural retention of participants, etc.?

Please provide more evidence that research capacity impacts satisfaction with career or that research capacity-building has a positive impact on healthcare workers’ satisfaction with their career. Right now, the paper states that there is a shortage, that practicing in rural areas is perceived as career-limiting and that the solution could be building research capacity but there is no evidence of why that would help. Why do you think this will have a strong impact on rural workforce recruitment and retention? Further, please explicitly make a connection of how building research capacity can help one’s career, especially when remaining in a rural area.

Line 137: Add “s” to “health professionalS”

Reviewer #2: The research question outlined is expected to address a valid academic problem or topic and contribute to the base of knowledge in the field.-----The authors have been very restrictive in the definition of their research question and while the outcome of retention is key to the healthcare workforce (and there is a wide literature in that area), it is unclear whether it is appropriate to just look at one type of retention enhancement focused on research (and research building capability) or whether its value needs to be considered relative to other retention strategies. Furthermore, given the lack of extensive evidence in the area of using research to enhance retention, it is unclear why the authors have chosen to focus narrowly on Australia in this project, particularly when retention issues are a global problem and there may be useful studies from other countries. It is unclear to me how many articles could be found to address the rather narrow research question as currently addressed in this paper. In my attempt to search on this topic for one of the databases mentioned, I came up with only 200 articles and most proved irrelevant (wrong location, wrong subject). It may be helpful for the authors to indicate what search terms will be used to capture sufficient articles on this topic.

Is the protocol technically sound and planned in a manner that will lead to a meaningful outcome and allow testing the stated hypotheses? It is unclear why a systematic review is needed at this time when there is little evidence on the existing scope of literature in this area. It would be helpful to hear whether the authors considered a scoping review first before choosing a systematic review. Or could they explain further why the systematic review is more useful than a scoping review here. It seems unlikely that "the relationship between health workforce engagement in research or research capability-building programs and the attraction and retention of health professionals in regional Australia" can be easily compared across different studies, if, as I would surmise, existing literature in this area provides exploratory rather than confirmatory evidence.

Overall, the authors have an interesting idea to identify ways that research and research-enhancing work supports a remote workforce, but their questions are too narrowly defined and they could learn more by comparing across countries.

**Do you want your identity to be public for this peer review?** For information about this choice, including consent withdrawal, please see our Privacy Policy

Reviewer #1: **Yes:** Kelley Arredondo

Reviewer #2: No

---

## [Author Response · Author response to Decision Letter 1]

25 Nov 2025

We sincerely appreciate the opportunity to revise our manuscript, "Engagement in research capability-building: Impact on healthcare workforce attraction and retention in rural and remote Australia – A scoping review protocol", for further consideration for publication in PLOS ONE. We are grateful for the thoughtful and constructive comments provided by the editor and reviewers, which will strengthen our actual scoping review work.

We have carefully addressed all comments and incorporated the suggested changes into the revised protocol. Furthermore, detailed point-by-point responses outlining the revisions are provided in the 'Response to reviewers' file.

Thank you.

---

## [Decision Letter · Decision Letter 1]

18 Dec 2025

Engagement in research capability-building: Impact on healthcare workforce attraction and retention in rural and remote Australia – A scoping review protocol

PONE-D-25-29995R1

Dear Dr. Haile,

We’re pleased to inform you that your manuscript has been judged scientifically suitable for publication and will be formally accepted for publication once it meets all outstanding technical requirements.

Kind regards,

Muhammad Shahzad Aslam, Ph.D.,M.Phil., Pharm-D

Academic Editor

PLOS One

Additional Editor Comments (optional):

Reviewers' comments:

Reviewer's Responses to Questions

**Comments to the Author**

1. Does the manuscript provide a valid rationale for the proposed study, with clearly identified and justified research questions?

Reviewer #1: Yes

2. Is the protocol technically sound and planned in a manner that will lead to a meaningful outcome and allow testing the stated hypotheses?

Reviewer #1: Yes

3. Is the methodology feasible and described in sufficient detail to allow the work to be replicable?

Reviewer #1: Yes

4. Have the authors described where all data underlying the findings will be made available when the study is complete?

Reviewer #1: Yes

5. Is the manuscript presented in an intelligible fashion and written in standard English?

Reviewer #1: Yes

You may also provide optional suggestions and comments to authors that they might find helpful in planning their study.

Reviewer #1: All comments were addressed appropriately by the authors in their response to reviewers for my comments

**Do you want your identity to be public for this peer review?** For information about this choice, including consent withdrawal, please see our Privacy Policy

Reviewer #1: No

---

## [Editor Report · Acceptance letter]

PONE-D-25-29995R1

PLOS One

Dear Dr. Haile,

I'm pleased to inform you that your manuscript has been deemed suitable for publication in PLOS One. Congratulations! Your manuscript is now being handed over to our production team.

Kind regards,

on behalf of

Dr. Muhammad Shahzad Aslam

Academic Editor

PLOS One